# Sustainable Home Meal Replacement (HMR) Consumption in Korea: Exploring Service Strategies Using a Modified Importance–Performance Analysis

**DOI:** 10.3390/foods11060889

**Published:** 2022-03-21

**Authors:** Yunho Ji, Jangheon Han

**Affiliations:** 1College of Business Administration, Kangwon National University, 1 Kangwondaehak-Gil, Chuncheon 24341, Korea; yunho.ji@kangwon.ac.kr; 2Humanitas College, Kyung Hee University, 26 Kyungheedae-ro, Dongdaemun-gu, Seoul 02447, Korea

**Keywords:** home meal replacement, modified IPA, sustainable consumption, service strategy, COVID-19

## Abstract

The COVID-19 pandemic has changed traditional consumer consumption behavior and requires a new service strategy to cope with sustainable consumption. Moreover, it is necessary to focus academic attention on consumer behavior to cook and eat more easily amid Korea’s socioeconomic changes such as the increase in single-person households, aging, rising prices, and continuing economic recession. In this study, we used a revised importance–performance analysis (IPA) to identify specific measures to improve consumer satisfaction with home meal replacements (HMRs). An online survey of Korean adults who had purchased HMRs was conducted based on a convenience sampling method. According to the results, items that could be intensively improved were the ‘introduction of new dishes’, ‘creation of various flavors’, ‘cost-effectiveness, and ‘reasonable price’, whereas ‘easy to prepare’, ‘easily available’, and ‘identified country of origin’ received favorable assessments. With the growth of the non-face-to-face economy due to COVID-19, our findings suggest marketing strategies for sustainable HMR consumption.

## 1. Introduction

The home meal replacement (HMR) market is growing rapidly due to the increased number of single-person households and women’s economic participation. Globally, major food manufacturers and supermarkets are actively developing HMRs to expand their markets [1,2]. Moreover, the COVID-19 pandemic led to restrictions on private gatherings and telecommuting; thus, increasing the demand for HMRs [3]. HMRs have also become more competitive, as consumers choose products based on price-to-performance assessments and convenience when purchasing food [4,5,6]. This trend is expected to continue after the COVID-19 pandemic has ended [7,8].

The consumer value of HMRs should reflect consumers’ emotional and empirical perceptions of the product [9,10]. From a marketing point of view, identifying consumer attitudes, purchasing motivations, and interest in food products’ functional efficiency are important considerations in the food-service sector [11]. Therefore, the first step in establishing a sustainable HMR market ecosystem is to identify the attributes of HMRs and their relation to consumer consumption behavior. In the ‘Usage Status and Improvement Plans of Domestic Raw Materials in the HMR Industry’, reported by the Korea Rural Institute [12], the HMR market in 2018 was valued at about 3.7 trillion KRW, with an average annual growth rate that had more than doubled between 2010 and 2018 to 16.1%. Moreover, HMRs have expanded and range from simple meal kits and ready meal products to a ready meal replacement (RMR). The restaurant industry, which COVID-19 has strongly hit, is seeking entry into the HMR market, such that producers find themselves competing with well-known restaurants and bakery brands [13]. This expanding market has caused confusion over the definition of HMRs and revealed a lack of reference data; thus, hindering market analyses.

What service strategies should the HMR market have for sustainable consumption? Academic research has explored HMRs from consumer and nutritional perspectives. Among the former, e.g., [14,15,16,17], are studies on consumer attitudes and behavioral causality, e.g., [5,18,19,20], as well as analyses of HMR markets and marketing strategies, e.g., [2,7,21]. Studies of HMRs from a nutritional perspective have mostly focused on the nutritional balance and quality of HMRs, e.g., [3,22,23]. Despite HMR-related research achievements in various academic fields, research to understand the characteristics of products is essential in seeking a sustainable service strategy of COVID-19 [24,25]. The IPA model is used as a significant model for evaluating service quality performance and predicting consumer behavior [26,27,28].

This study aimed to identify features that make HMRs sold in South Korea attractive to Korean consumers. As a specific research method, the concept and characteristics of HMRs were identified, and the current status and trends of the HMR market were investigated. Specifically, we conducted a revised importance–performance analysis (IPA) to determine the measures to improve consumer satisfaction with HMRs. As the non-face-to-face economy assumes increasing importance, our results offer marketing strategies necessary for a sustainable HMR consumption.

## 2. Review of the Literature

### 2.1. HMR Characteristics and Consumptions

HMRs consist of either main dishes or the critical components for a dish offered in single or multiple packaging containers. The meals contain protein, carbohydrates, and vegetable sources, and are designed to save time in preparing home-cooked meals [29]. Thus, HMRs include both fully and semi-cooked home-style products for immediate consumption or after simple preparation [14]. HMRs are time-savers for people seeking convenience and high-quality meals [19]. A significant advantage of HMRs is that difficult-to-cook dishes can be simplified, such that they can be easily enjoyed at home at a low cost [16,30]. This time-saving aspect is key to the growth potential of HMRs [21], together with current food culture trends that emphasize organic sources, healthy food, convenience, and small purchases to limit waste, all of which have had a significant impact on the popularity and growing interest in HMRs [6].

Modern people’s dietary lifestyle is much simpler and faster than before [31,32]. In particular, this tendency is stronger for Asians than for Westerners and tends to pursue convenience in the traditional diet [33]. Additionally, the spread of food delivery culture in pandemics led to the growth of HMRs [34]. The changed dietary lifestyle can explain and predict HMR consumption behavior as food purchase behavior [35,36]. Wise consumers seek institutional, scientific, and biologically safe foods [37]. Previously, consumers mainly purchased HMRs at large discount stores but purchased more online after COVID-19. HMR marketing activities are also primarily carried out on SNS and online to target the Z generation and young couples who are highly dependent on mobile purchases [38,39]. Since an HMR is a consumer product with a relatively low degree of involvement, innovative supply management, such as an e-chain system or quality certification, can significantly influence decision making [40].

HMRs have been referred to as ‘instant food’, ‘delivery food’, ‘simple food’, ‘packaged food’, and ‘takeout’, depending on the perspective and context of the conversation, but a broadly accepted definition for the industry is lacking [41]. Nevertheless, HMRs have steadily developed to meet consumer needs due to developments in the production and distribution technology. Based on the Duct National Food Tour 1997–1998 database, Costa et al. [29] classified HMRs into four categories as summarized in Table 1: ready to eat (RTE), ready to heat (RTH), ready to cook (RTC), and ready to prepare (RTP).

### 2.2. Attributes of HMRs

Attributes provide insights into consumers’ attitudes and consumption propensities when purchasing a product. They distinguish a product from competing products and refer to the tangible and intangible features of the product [42]. Current studies of the attributes of HMRs take into account the diverse needs of consumers and the diversification of distribution channels, such as online [43,44]. Consumer attitudes toward the attributes of HMRs have become more critical, especially as food has shifted to online distribution [45]. These economic and social changes cause lifestyle seeking simplicity, convenience, and practicality to be common [46].

With the expansion of the HMR market and the launch of numerous HMR products, consumers have increasingly opted for HMRs as an alternative to eating out. A dietary lifestyle measures consumer behavior that reflects consumers’ life values with external factors such as social perspectives, traditions, and safety. Internal factors measure the consumption behavior of the product, such as health, hygiene, taste, product quality, freshness, convenience, and information about the product [47]. HMR attributes are generally determined by the consumer’s food-related lifestyle and food consumption patterns [48], although the latter may also be influenced by the available distribution channels [49]. Creed [50] stressed that consumers who eat out often are more aware of the importance of HMR manufacturing processes and methods, such that food companies should establish a sustainable manufacturing and distribution system of HMRs. This can be accomplished through the development of innovative technologies that also take into account consumers’ interests in nutrition, convenience, and safety.

Figiel and Kufel [51] emphasized five elements of global food market R&D trends: health, pleasure, physical, convenience, and ethics. HMRs comprise a newly developed food industry group designed for busy modern people seeking time-saving measures and efficiency. Choi et al. [20] noted that in developing countries such as Vietnam, contextual values are often reflected in consumption trends. In this context, emphasizing the shelf life of HMRs, hygiene, nutrition, and other core attributes is essential in a culture that values food safety. Jang et al. [16] recognized changes in consumers’ dietary lifestyles as a significant trend in sustainable consumption, including the increasing proportion of older adults in many countries. For this segment of the population, HMR attributes such as food quality, menu, price, reliability, and delivery service quality are essential motivations for the consumer and a significant contributor to market segmentation. Cho et al. [17] listed food quality and menu diversity as core attributes capable of satisfying consumer aesthetic and functional needs, but health-oriented, convenience, and price concerns were also important. Since COVID-19, HMR packaging technology has advanced in response to consumer concerns. These advances include active packaging technologies, such as a modified atmosphere and antimicrobial packaging, extending product shelf life, maintaining product quality, and preserving freshness [50].

### 2.3. The HMR Market

The growth of HMRs has been attributed to the increase in single-person households, the increased participation of women in the economy, increased leisure activities (travel), and the loss of restaurant dining opportunities due to COVID-19 [23]. In parallel, the concept of well-being has expanded such that it now encompasses not only physical health, but also the pursuit of pleasure and the sharing of experiences, both of which have significantly impacted food culture [52].

While the HMR market shares many features with the food processing industry, it is susceptible to changes in current trends in a rapidly growing and increasingly competing [15,53]. Thus, HMR marketing enterprises must be aware of changes in consumer preferences and can develop new products as competition intensifies. In addition, opportunities for small and medium-sized enterprises (SMEs) to participate in the market have expanded due to changes in distribution channels, especially the online market. Similarly, the expansion of the overseas market has been facilitated by the globalization of food culture and improved food storage technology and logistics [54]. According to the Korean Ministry of Food and Drug Safety’s annual report [55,56,57,58,59], ‘Production performance of food and food additives’, the HMR market grew by an average of 15.2% annually, reaching 4 trillion and 42.5 billion KRW, respectively, in 2020. As of 2020, RTC accounted for 45.4%, RTE for 32.0%, RTP for 17.5%, and RTH for 5.0% of the Korean market. The year-to-year growth rate of RTH was the highest (21.1%), followed equally by RTC (11.8%) and RTE (11.8%).

With the expansion of online food shopping, morning delivery, in which groceries ordered at night arrive at the purchaser’s door early the following day, has become a new HMR distribution structure [60]. Global distribution and logistics companies such as Amazon have recently been competitively expanding their investment in the cold chain to more quickly deliver the fresh food they need at a small quantity and reasonable cost [61]. According to DeepSearch, which specializes in data collection on Korean trends, the morning delivery market uses the cold chain, i.e., technology and management techniques developed for temperature- and humidity-sensitive products, such as food and medicine, within the global supply chain [62], which grew from 10 billion KRW in 2015 to 8000 billion KRW in 2019 [63]. Reliance on the cold chain is continuously expanding as the demand for better food quality and more sustainable logistics, including less waste and reduced energy usage, grows [64,65]. Thus, when HMRs became a growth engine for the food industry, they also stimulated research into food sustainability (Table 2).

## 3. Research Method

### 3.1. Questionnaire Design

In this study, a modified IPA was used to assess the importance to consumers of the various attributes of HMRs related to sustainable consumption. An initial questionnaire based on prior studies of food services [14,15,16,17] was drafted for use in quantitative analysis. The suitability and applicability of the questionnaire were reviewed in a preliminary survey, after which the questionnaire was accordingly revised and supplemented. Six questions, four addressing demographics and two on the HMR consumption characteristics of the respondents, were assessed using nominal measures. The importance of HMRs, consumer satisfaction, and overall satisfaction was addressed in 16 questions according to a Likert 5-point scale.

### 3.2. Data Collection and Analysis

The survey data of this study were collected in South Korea from 12 to 25 May 2020, for adults who purchased HMRs for the past three months. The survey was conducted online using a convenience sampling method. From the 344 questionnaires that were collected, 306 were used in the final analysis; the 38 excluded questionnaires contained unanswered questions or invalid responses.

The questionnaires were first evaluated in a frequency analysis to summarize respondent demographics and consumption characteristics. Statistical analysis was then conducted to obtain the average value of each item and, thus, to rank the queried items. This was followed by an IPA, as revised by Deng [26], performed using Excel and SPSS 24.0. The revised IPA calculated the implicitly derived importance of and satisfaction with specific attributes, in this study being the HMRs based on a natural logarithm (ln) and partial correlation coefficients (PCCs).

## 4. Results

### 4.1. Characteristics of the Respondents

The sex and age distribution of the respondents, their employment status, monthly income, and HMR consumption characteristics, including the number of monthly purchases and the purchase cost of HMRs, are displayed in Table 3.

### 4.2. Perceived Satisfaction of HMR Attributes

The average and standard deviation of 16 satisfaction levels related to attributes of HMR were calculated. The average score ranking of 306 respondents’ satisfaction with the HMR attributes is shown in Table 4. The mean scores for all 16 satisfaction attributes ranged from a low of 2.7148 to a high of 4.3816; all 16 satisfaction attributes had standard deviations of less than 1.006. The average and standard deviation of overall satisfaction with the HMR attribute were 3.65 points and 0.68 points, respectively. Moreover, the satisfaction ranking of the HMR attributes was based on the ‘average’ score.

Consequently, a high level of satisfaction was found for items related to HMR convenience, such as ‘simple to cook (4.3816)’, ‘readily available (4.2623)’, ‘reasonably priced (4.2557)’, and ‘easily stored (4.2000)’. On the other hand, the lowest-ranked items with low satisfaction were three items: ‘Unique Taste (2.7148)’, ‘New Meal Plan (2.9148)’, and ‘Clean Packaging (3.1344)’, which needed to be improved to enhance consumer satisfaction.

### 4.3. Implicitly Derived Importance of HMR Attributes

The revised IPA presented by Deng [26] converts the data for each property into a natural logarithm and then derives PCCs; thus, generating a critical value for each property. In the traditional IPA, linear results are also obtained in areas in which the importance of/satisfaction with an attribute is high or low. The modified IPA has been applied to many studies related to consumer behavior and marketing to better explain the exact degree of consumer awareness [27,28].

The implicitly derived importance of the attributes ranges between 0.2421 and 0.0049; it was also ranked according to value. According to the modified IPA, the implicitly derived importance (expressed by PCC reported in brackets) of HMR attributes was: ‘easy to prepare (0.182)’, ‘introduction of new dishes (0.165)’, ‘cost-effective (0.110)’, and ‘country of origin (0.088)’. This result shows that all of the potentially significant attributes of HMRs cited in the Introduction, i.e., taste, price, convenience, and hygiene, were identified as being of implicitly derived importance. Items with a low implicitly derived importance were: ‘accurate information on the ingredients (−0.007)’, ‘new flavor (−0.006)’, and ‘simple to cook (−0.038)’. A summary of the attribute analysis is presented in Table 5.

### 4.4. Modified IPA Results

To derive the IPA matrix, satisfaction was set as the x-axis and the implicitly derived importance as the y-axis (Table 6). Each mean of the determined implicitly derived importance and satisfaction level was assigned as the grid’s intermediate standards, and the results were derived as shown in Figure 1.

Quadrant I consists of items of low satisfaction but high importance; thus, indicating where urgent efforts, rapid action, and modifications are required. Quadrant II consists of items of high satisfaction and importance, indicating that the current situation should be maintained. Quadrant III is composed of elements with low satisfaction and importance and, thus, of relatively low priority. Quadrant IV items are of low relative importance but high satisfaction, so they can be maintained to some extent without further effort.

## 5. Discussion

In this study, the importance of and satisfaction with HMR products for consumers were empirically examined using a revised IPA. The primary data obtained in the analysis can be applied to product development and brand marketing strategies for sustainable HMR consumption in the food industry. Many studies using the IPA have proposed recommendations of service improvement and marketing strategies based on Quadrant I (concentrate here) and Quadrant II (keep up the good work) [66,67]. Since consumption value is often perceived through individual consumption experience [10], the results derived from our IPA could be of value to the HMR industry. These results can be summarized as follows.

First, four HMR attributes should be intensively improved: ‘introduction of new dishes’, ‘creation of various flavors’, ‘cost-effectiveness, and ‘reasonable price’. These items correspond to the HMR attributes of quality and price. They are also the most competitive items, and those that HMR producers tend to copy, rather than work to develop new dishes for their menus. Therefore, in-depth R&D is needed to consider the various requirements of HMR consumers.

The creation of the taste of HMRs should be considered not only in the social and cultural environment of the market, but also in the aesthetic and moral decisions of consumers. As global interest in health and the environment increases, the consumption of healthy foods such as vegetables and organic foods increases [68]. HMRs should develop premium organic menus that consider vegetarian diets, nutritional selection attributes, and purchasing attitudes. Today’s elderly people are defined as new generations, and accordingly, new perspectives and preparations are needed. The most fundamental concern of these generations, whose assets have increased and tastes have diversified, is “health care.” Moreover, the starting point of their health care always starts with food. Therefore, R&D of HMR for active-aging should be preceded, and brand development and marketing strategies that take into account their diet and activity patterns are needed.

Items corresponding to the price of HMRs were among those requiring intensive improvement. As there are more alternatives that consumers can choose, consumers react sensitively to prices and value cost-effectiveness [20]. The rising consumer Z generation shows more rational and ethical consumption tendencies than in the past [69,70]. They aim for empirical consumption, not material consumption; they buy the necessary items to a minimum or wander for a large discount even though they can afford it economically [71]. On the other hand, eco-friendly products strive for sustainable consumption at a little more cost. In other words, it is necessary to develop an HMR that can save time and use it practically. It is necessary to find attributes that affect consumer price sensitivity and induce consumers to recognize products and consume reasonably. In addition, it is necessary to develop manufacturing methods and distribution processes that increase price competitiveness. Food companies should closely examine the diet, lifestyle, and food preferences of HMR users to reduce purchasing and raw material costs.

Second, the factors ‘easy to prepare’, ‘easily available’, and ‘country of origin’ were cited as those not requiring further improvement (keep up the good work). Today, consumers seek time-saving tools and convenience, while also considering health and nutrition. This includes new, high-quality meal solutions that offer an experience beyond the routine of eating [21]. The attributes of HMRs shape consumers’ relative preferences and change their consumption behavior [18,72]. Therefore, food companies should implement marketing strategies that consider consumers’ financial and time resources. SNS marketing is helpful for brand positioning for customers by producing content that promotes brands or products [73]. It is good to open cooking classes and introduce new foods through YouTube or SNS. Providing tips to demonstrate how to cook HMRs simply or eat them deliciously could help consumers recognize them as friendly and willing to purchase them. In addition to indicating the country of origin on the packaging surface of HMR products, information on their nutritional content, convenience, etc., should be included, according to the Foreign Trade Act standards. In preparation for the post-COVID-19 era, food companies should build a cold-chain system to ensure the hygiene of HMR products and, thus, build trust in encouraging consumers’ sustainable consumption. Suppliers should establish their energy efficiency standards and service standards to produce, store stably, and distribute products through cold chains.

## 6. Conclusions

Amid socio-economic changes, such as rising prices, a continued economic recession, an increase in single-person households, and aging populations, consumer interest in convenient but sustainable eating has grown, and HMR services have been increasingly utilized.

Since COVID-19 has prevented consumers from freely eating out, there is currently an excellent opportunity to benchmark good-quality restaurant dishes and commercialize them as HMRs. For restaurants that now offer HMRs, it is vital to commercialize popular dishes such that their quality is similar to that of the original versions. It should also be noted that with the enormous amount of food-related information available in the mass media, interest in heathy food and diet has increased. Therefore, it is necessary to develop high-quality HMR products that can support consumers’ dietary habits and lifestyles and marketing strategies for each market segment. As COVID-19 has ruled out many traditional distribution methods by limiting physical access to customers, social network sites and online open markets should be easily accessible to the public and serve as channels to introduce new products to customers.

Another condition that consumers value in decision making is sustainability [74]. The number of pollutants emitted in the process of product production or service execution, recyclability after use, and whether toxic chemicals have been replaced with natural ingredients have become essential factors in consumers’ purchasing decisions [75]. In addition, food upcycling is emerging as a new trend. Eco-friendly products are being developed created from food by-products that were naturally discarded, such as bean curd and broken rice. Therefore, there is a need for continuous R&D that can satisfy both sustainability and value consumption. In addition, innovative solutions should be developed along with a conscious supply chain, and responsible ESG management should be introduced so that the HMR market system can be transformed into a sustainable way.

Despite this study’s theoretical and practical implications, the following limitations should be noted. First, since only the Korean market was included in the survey, our results may not represent consumer behavior based on global consumption trends or the consumption patterns of other countries. Second, since the survey was conducted during the COVID-19 pandemic, HMR consumption behavior may not be the same after the pandemic. Third, implications for all HMR items in the fourth quadrant of IPA results were not derived, and consumer behavior control and impact variables were not considered. The attributes of HMR are expected to become more diverse as the supply system continues to develop and the food market becomes globalized. Future research should identify problems of HMR consumption from a consumer perspective and find ways to improve services. Additionally, from the supplier’s point of view, an HMR supply system improvement plan and a sustainable industrial growth plan should be prepared.

## Figures and Tables

**Figure 1 foods-11-00889-f001:**
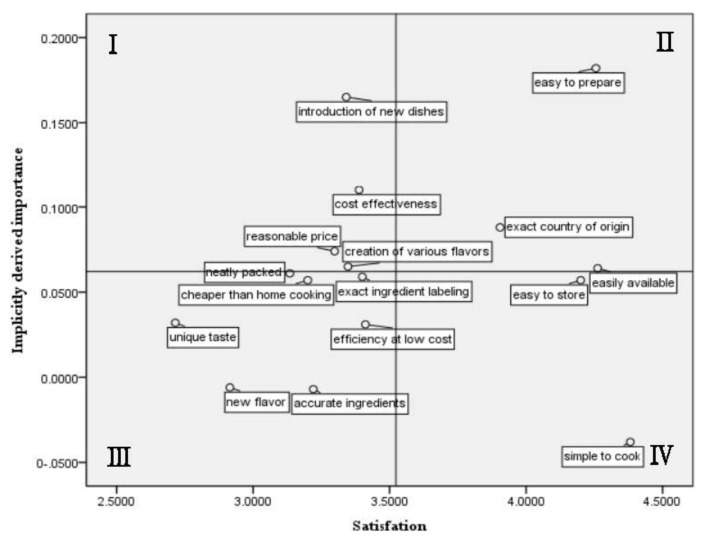
Matrix results of an importance–performance analysis of home meal replacement attributes.

**Table 1 foods-11-00889-t001:** Classifications and descriptions of home meal replacements (HMRs).

Classification	Definition
Ready to eat (RTE)	Can be consumed immediately after purchase, without any cooking, e.g., lunch boxes, hamburgers, sandwiches, and salads
Ready to heat (RTH)	Require brief heating in a microwave or hot water, e.g., instant spaghetti or curry, instant rice, and soup
Ready to cook (RTC)	Require cooking utensils (e.g., frying pans, pots, and ovens), a relatively long heating time or a simple cooking process, e.g., pork tripe, steak, frozen fried rice and soup, stew, and stir-fried and steamed foods
Ready to prepare (RTP)	Contain small portions, fresh ingredients, and sauces, and can be consumed after a series of cooking processes, e.g., stew kit, herb kit, steak kit, and steam kit

Reproduced with permission from [29]; published by Elsevier Science, 2001.

**Table 2 foods-11-00889-t002:** Annual HMR sales in Korea (in KRW).

Classification	2016	2017	2018	2019	2020	Annual Average Growth (%)
RTE	652,340	800,972	1,530,521	1,583,074	1,416,162	25.2
RTH	109,959	163,041	181,734	184,531	224,639	19.6
RTC	1,330,585	1,767,897	1,317,778	1,694,898	2,010,327	15.2
RTP	705,262	664,459	678,034	743,403	774,251	12.1
Total	2,798,147	3,396,370	3,708,068	4,205,908	4,425,381	15.2

Note: The numbers are 1 million KRW units. Reproduced with permission from [55,56,57,58,59]; Korean statistical Information Service (https://kosis.kr/index/index.do, 15 February 2022).

**Table 3 foods-11-00889-t003:** Demographics and consumption characteristics of the respondents.

Variables		Frq.	%
Sex	Male	140	45.8
	Female	166	54.2
Age (years)	<30	32	10.5
	30–39	137	44.8
	40–49	71	23.2
	≥50	66	21.6
Job	Students	11	3.6
	Office	106	34.6
	Service	33	10.8
	Housewife	77	25.2
	Self-employed	36	11.8
	Others	43	14.1
Monthly income	<$2000	48	15.7
	$2000–$3000	90	29.4
	$3000–$5000	72	23.5
	>$5000	96	31.4
HMR usage frequency	0–2 times	124	40.5
	3–5 times	93	30.4
	6–9 times	40	13.1
	≥10 times	49	16
HMR expenditures	<$6	202	62
	$6–$10	68	22.2
	≥$10	36	11.8
Total	306		

**Table 4 foods-11-00889-t004:** The satisfaction results of HMR attributes.

Customer Attribute	Satisfaction		
	Mean	Standard Deviation	Rank
unique taste	2.7148	1.00686	16
creation of various flavors	3.3475	0.94792	9
introduction of new dishes	3.3410	0.97424	10
new flavor	2.9148	0.94204	15
cheaper than home cooking	3.2000	1.03364	13
reasonable price	3.2984	0.85811	11
efficiency at low cost	3.4112	0.91125	6
cost-effectiveness	3.3882	0.90506	8
easy to prepare	4.2557	0.63363	3
simple to cook	4.3816	0.70298	1
easy to store	4.2000	0.78388	4
easily available	4.2623	0.72324	2
neatly packed	3.1344	0.83408	14
accurate ingredient information	3.2204	0.87547	12
exact ingredient labeling	3.4000	0.83745	7
exact country of origin	3.9046	0.82501	5
average	3.5234	-	

**Table 5 foods-11-00889-t005:** Implicitly derived importance of HMR attributes.

Customer Attribute	Implicitly Derived Importance			
	PCC	(ln) Satisfaction	Standard Deviation	Rank
unique taste	0.032	0.9212	0.41253	12
creation of various flavors	0.065	1.1586	0.33734	6
introduction of new dishes	0.165	1.1578	0.32478	2
new flavor	−0.006	1.0142	0.34366	15
cheaper than home cooking	0.057	1.0997	0.37912	10(2)
reasonable price	0.074	1.1523	0.30559	5
efficiency at low cost	0.031	1.1818	0.32381	13
cost-effectiveness	0.110	1.1763	0.31783	3
easy to prepare	0.182	1.4364	0.15751	1
simple to cook	−0.038	1.4616	0.19094	14
easy to store	0.057	1.4144	0.21421	10(2)
easily available	0.064	1.4330	0.19213	7
neatly packed	0.061	1.1010	0.30453	8
accurate ingredient information	−0.007	1.1281	0.29930	16
exact ingredient labeling	0.059	1.1883	0.28071	9
the exact country of origin	0.088	1.3375	0.22962	4
average	0.062			

Note: PCC—partial correlation coefficients; ln—natural logarithm.

**Table 6 foods-11-00889-t006:** Results of the modified IPA.

Extracted Dimension	Customer Attribute	Satisfaction	Implicitly Derived Importance
I (concentrate here)	introduction of new dishes	3.3410	0.165
	cost-effectiveness	3.3882	0.110
	reasonable price	3.2984	0.074
	creation of various flavors	3.3475	0.065
II (keep up the good work)	easy to prepare	4.2557	0.182
	the exact country of origin	3.9046	0.088
	easily available	4.2623	0.064
III (low priority)	cheaper than home cooking	3.2000	0.057
	unique taste	2.7148	0.032
	neatly packed	3.1344	0.061
	exact ingredient labeling	3.4000	0.059
	efficiency at low cost	3.4112	0.031
	new flavor	2.9148	−0.006
	accurate ingredients	3.2204	−0.007
IV (possible overkill)	easy to store	4.2000	0.057
	simple to cook	4.3816	−0.038

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
