# Peer review of "Sustainable Home Meal Replacement (HMR) Consumption in Korea: Exploring Service Strategies Using a Modified Importance–Performance Analysis"

_foods, 2022, doi:10.3390/foods11060889_

Round 1

Reviewer 1 Report

The paper deals with an interesting topic in the fields of food services and food habits. Some enhancements can furtherly stress the importance of the results. Suggestions are reported below.

  • It would be better to cite Korea in the title
  • Introduction
    • Please, add references for the modified IPA approach at lines 63
  • Research methods
    • I would suggest compacting the modifications made on the Importance-Performance Analysis method partly cited in the 3.Research methods ("The revised IPA overcomes the limitations of the traditional IPA by calculating the implicitly derived importance of and satisfaction with certain attributes- in this study, those of HMRs-based on a natural logarithm (ln) and partial correlation coefficients (PCCs). and partly in section 4.Results, particularly in 4.2 section. It is suggested to move all the methodological aspects in section 3.
    • it is not clear to me if the modifications to IPA were done as illustrated in Deng, 2007 or modifications have been made specifically for the present study. 
  • 4.Results
    • If I have understood well the 4.1 sub-section refers to traditional IPA. Is it correct? If not please, add some words to stress traditional vs. modified.
    • Lines 197-199 Please, add "PCC =" in the brackets where figures are reported. Moreover, the values are very close to 0. In my opinion, this means that the dimensions are not relevant at all. Am I wrong?
    • Figure 1 I would suggest adding titles reporting the quadrant to match the figure and the text.
    • Lines 211-213 “Many studies using the IPA have made recommendations of service improvement and marketing strategies based on Quadrant I (concentrate here) and Quadrant II (keep up the good work)” seem more related to a discussion (please, move to the discussion and add references)
    • Do the Authors mean that being of “Low Importance items” is not to be considered by foodservice and food industry? If yes, please add to the discussion and align the conclusions.
    • Lines 214-218 “Ultimately, Quadrant I items (introduction of new dishes, new meal planning, cost-effectiveness, and reasonable price), need to be intensively improved and a strategy accordingly designed. Quadrant II's components (easy to prepare, easily available, and exact country of origin) need to be continuously managed and controlled to maintain the current state of sustainable consumption” seem more conclusions than results.
  • Conclusions and implications
    • I would consider widening the section by developing a discussion comparing the results with previous studies and justifying the conclusions about the needs to intensively improve four HMR attributes. In case I would split “discussion” and “conclusions”.
    • Please, add references to the blocks of lines 229-241, 242-245, 246-249, 263-271.
    • Conclusions are not clear. Is it possible that conclusions are reported at lines 214-218 when discussing the quadrants resulting from the modified IPA application?

Editing

  • Table 1. It is suggested to replace the abbreviations with extended wording.
  • Table 3. Please add footnotes explaining the abbreviations.

Author Response

The paper deals with an interesting topic in the fields of food services and food habits. Some enhancements can furtherly stress the importance of the results. Suggestions are reported below.

Response: Our research team is trying to make good results by carefully examining your evaluation and suggestions. Thank you very much in advance for your advice.

  • It would be better to cite Korea in the title

Response: Thank you for your suggestion. However, we would like to maintain the existing title to understand the concept and characteristics of HMR from a broader perspective. Please understand that.

  •  
  •  
  • Introduction
    • Please, add references for the modified IPA approach at lines 63

Response: Per your suggestion, While attaching references as follows, the importance of IPA analysis was emphasized and the basis for the implementation of the analysis was revealed to indicate the differences in research methodology in this study. Please see line numbers from 56 to 60 on page 2. Thank you.

“Despite HMR-related research achievements in various academic fields, research to understand the characteristics of products is essential in seeking a sustainable service strategy of COVID-19 [25, 25]. The IPA model is used as a significant model for evaluating service quality performance and predicting consumer behavior [26, 27, 28].”

  • Research methods
    • I would suggest compacting the modifications made on the Importance-Performance Analysis method partly cited in the 3.Research methods ("The revised IPA overcomes the limitations of the traditional IPA by calculating the implicitly derived importance of and satisfaction with certain attributes- in this study, those of HMRs-based on a natural logarithm (ln) and partial correlation coefficients (PCCs). and partly in section 4.Results, particularly in 4.2 section. It is suggested to move all the methodological aspects in section 3.

Response: In accordance with your suggestion, the methodological explanation was briefly explained and refined. Please see line numbers from 195 to 197 on page 4-5.

Meanwhile, in the case of your section integration proposal, instead of combining sections, we tried to explain 4.2 sections in more detail and provide richer meaning for the natural flow of this study. Therefore, the existing Table 3 is divided into 'Table 4. Situation' and 'Table 5. Impulsively derived importance of HMR attributes', and the IPA analysis process is explained in detail. Thank you for pointing out the issue. We hope this modification adequately addresses your comments. Thank you.

“The revised IPA calculates the implicitly derived importance of and satisfaction with specific attributes- in this study, those of HMRs-based on a natural logarithm (ln) and partial correlation coefficients (PCCs).”

  • it is not clear to me if the modifications to IPA were done as illustrated in Deng, 2007 or modifications have been made specifically for the present study. 

Response: According to your suggestion, the reason for implementing IPA in this study was presented in the introduction, and the validity and effectiveness of IPA used in previous studies were presented as a basis along with references. Please see line numbers from 58 to 60 on page 2, and from 225 to 227 on page 7. Thank you for your reasonable opinion.

“The IPA model is used as a significant model for evaluating service quality performance and predicting consumer behavior [26, 27, 28].”

4.3 Implicitly derived importance of HMR attributes

“(Modified) IPA has been applied to many studies related to consumer behavior and marketing to explain better the exact degree of consumer awareness [27, 28]. “

  • Results
    • If I have understood well the 4.1 sub-section refers to traditional IPA. Is it correct? If not please, add some words to stress traditional vs. modified.
    • Lines 197-199 Please, add "PCC =" in the brackets where figures are reported. Moreover, the values are very close to 0. In my opinion, this means that the dimensions are not relevant at all. Am I wrong?

Response: In this study, traditional IPA was not specifically presented. The reason is that many previous studies have proved the superiority of the revised IPA.

Therefore, in this study, to maximize the differentiation of the modified IPA, it was presented by dividing it into Table 4 and Table 5. In addition, in order to reduce confusion, it was explained that the PCC value was used as an implicitly derived importance value. Please see line numbers from 228 to 230 on page 7. We hope this addition adequately addresses your comments. Thank you.

“The implicitly derived importance of the attributes ranges between 0.2421 and 0.0049, it is also ranked according to value. According to the modified IPA, implicitly derived importance(=PCC) of HMR attributes were:”

  •  
  •  
  • Figure 1 I would suggest adding titles reporting the quadrant to match the figure and the text.

Response: I revised it as you suggested. Thank you.

  • Lines 211-213 “Many studies using the IPA have made recommendations of service improvement and marketing strategies based on Quadrant I (concentrate here) and Quadrant II (keep up the good work)” seem more related to a discussion (please, move to the discussion and add references)
  • Response: As you suggested, we moved that part to discussion and added a reference. Please see line numbers from 259 to 261 on page 10. Thank you.

“Many studies using the IPA have made recommendations of service improvement and marketing strategies based on Quadrant I (concentrate here) and Quadrant II (keep up the good work) [66, 67].”

  • Do the Authors mean that being of “Low Importance items” is not to be considered by foodservice and food industry? If yes, please add to the discussion and align the conclusions.

Response:  Based on IPA results, we focused on practical strategies to respond to the sustainable consumption of consumers. While agreeing with the parts, you pointed out, the limitations of this study were described. In addition, it was proposed that the limitations of one such study could be supplemented in future studies. Please see line numbers from 345 to 351 on page 12. We hope this modification adequately addresses your comments. Thank you.

“Third, implications for all HMR items in the fourth quadrant of IPA results were not derived, and consumer behavior control and impact variables were not considered. The attributes of HMR are expected to become more diverse as the supply system continues to develop and the food market becomes globalized. Future research should identify problems of HMR consumption from a consumer perspective and find ways to improve services. Also, from the supplier's point of view, HMR's supply system improvement plan and sustainable industrial growth plan should be prepared.”

  • Lines 214-218 “Ultimately, Quadrant I items (introduction of new dishes, new meal planning, cost-effectiveness, and reasonable price), need to be intensively improved and a strategy accordingly designed. Quadrant II's components (easy to prepare, easily available, and exact country of origin) need to be continuously managed and controlled to maintain the current state of sustainable consumption” seem more conclusions than results.

Response:  As you suggested, the sentence corresponding to the conclusion was deleted. Instead, theoretical and practical implications were greatly strengthened in conclusion.

  •  
  • Conclusions and implications
    • I would consider widening the section by developing a discussion comparing the results with previous studies and justifying the conclusions about the needs to intensively improve four HMR attributes. In case I would split “5. Discussion” and “6. conclusions”.

Response:  As you suggested, it was divided into discussion and conclusion. Thank you for posting a reasonable opinion.

.

  • Please, add references to the blocks of lines 229-241, 242-245, 246-249, 263-271.

Response:  For the points you pointed out earlier, we tried to sufficiently present the [intensive improvement] and [continuous effort] parts of food service along with the latest consumer decision-making and purchase trends. In particular, the distribution and sales strategy of food services according to the tendency to purchase Generation z was supplemented from a sustainable consumption perspective. The following are sentences that add implications to the discussion. Please see line numbers from 271to 291, from 301 to 305, and 310 to 312 on page 11. And We hope this addition adequately addresses your comments. Thank you.

“The creation of the taste of HMR should be considered not only in the social and cultural environment of the market but also in the aesthetic and moral decisions of consumers. As global interest in health and the environment increases, consumption of healthy foods such as vegetables and organic foods increases [68]. HMR should develop premium organic menus that consider vegetarian diets, nutritional selection attributes, and purchasing attitudes. Today's elderly people are defined as new generations, and accordingly, new perspectives and preparations are needed. The most fundamental concern of these generations, whose assets have increased and tastes have diversified, is "health care." Moreover, the starting point of their health care always starts with food. If HMR develops it as an eco-friendly organic food, it can remove the image of instant food and do brand positioning as a sustainable food.

Items corresponding to the price of HMRs were among those requiring intensive improvement. As there are more alternatives that consumers can choose, consumers react sensitively to prices and value cost-effectiveness [20]. The rising consumer Z generation shows more rational and ethical consumption tendencies than in the past [69, 70]. They aim for empirical consumption, not material consumption; they buy the necessary items to a minimum or wander for a large discount even though they can afford it economically [71]. On the other hand, eco-friendly products strive for sustainable consumption at a little more cost. In other words, it is necessary to develop an HMR that can save time and use it practically. It is necessary to find attributes that affect consumer price sensitivity and induce consumers to recognize products and consume reasonably.”

“SNS marketing is helpful for brand positioning for customers by producing content that promotes brands or products [73]. It is good to open cooking classes and introduce new foods through YouTube or SNS. Providing tips to demonstrate how to cook HMR simply or eat it deliciously will help consumers recognize it as friendly and be willing to purchase it.”

“Suppliers should establish their energy efficiency standards and service standards to produce, store stably, and distribute products through cold chains.”

  • Conclusions are not clear. Is it possible that conclusions are reported at lines 214-218 when discussing the quadrants resulting from the modified IPA application?

Response:  Based on the results derived through empirical analysis as an extension of the above answer, we tried to present the future strategy of food service by focusing on sustainable consumption. Many discussions were dealt with in the discussion of 6. 7. In the conclusion, the main points of responding to COVID-19 and targeting eco-friendly markets were presented. Please see line numbers from 329 to 338 on page 12. We hope this addition adequately addressed your concern. Thank you.

“Another condition that consumers value in decision-making is sustainability [74]. The number of pollutants emitted in the process of product production or service execution, recyclability after use, and whether toxic chemicals have been replaced with natural ingredients have become essential factors in consumers' purchasing decisions [75]. In addition, food upcycling is emerging as a new trend. Eco-friendly products are being developed made from food by-products that were naturally discarded, such as bean curd and broken rice. Therefore, there is a need for continuous R&D that can satisfy both sustainability and value consumption. In addition, innovative solutions should be developed along with a conscious supply chain, and responsible ESG management should be introduced so that the HMR market system can be transformed into a sustainable way.”

Editing

  • Table 1. It is suggested to replace the abbreviations with extended wording.

Response:  Focusing on delivering the contents briefly. Thank you, though.

  • Table 3. Please add footnotes explaining the abbreviations.

Response:  I added a footnote as you suggested.

Response:  Most of the criticisms have helped this paper achieve completeness. Our research teams have learned a lot by contemplating and revising your point, and thank you once again for your active process support from MDPI.

With Omicron still actively spreading, I hope you will always be healthy.

Reviewer 2 Report

The paper covers a challenging topic of certain interest for researchers and academics.

The authors should consider the following recommendations in order to improve the original manuscript:

    • To include the structure of the paper in the Introduction section;
    • To include the research hypotheses;
    • There are no research questions so must be included;
    • There provided very few comments about COVID-19 pandemic and its globally devastating impact. How affects Covid-19 pandemic this research idea and its findings?
    • It is more than necessary to expand considerably the section "Literature review". The authors also did not provide sufficient evidence on literature review to support the hypotheses. The Literature review section which is practically non-existent being mentioned only a few bibliographic references quite uncorrelated. Authors should take into consideration much more recent publications in the sphere of discussed subject matter, especially studies conducted during the last 5 years.
    • Regarding consumer behaviour framework, I suggest extending the literature section by including at least the following relevant studies:

a) Hawaldar, I.T.; Ullal, M.S.; Birau, F.R.; Spulbar, C.M. Trapping Fake Discounts as Drivers of Real Revenues and Their Impact on Consumer’s Behavior in India: A Case Study. Sustainability 2019.

b) Timpanaro, G.; Bellia, C.; Foti, V.T.; Scuderi, A. Consumer Behaviour of Purchasing Biofortified Food Products. Sustainability 2020, 12, 6297.

c) Jing, X.; Guanxin, Y.; Panqian, D. Quality Decision-Making Behavior of Bodies Participating in the Agri-Foods E-Supply Chain. Sustainability 2020, 12, 1874.

    • Authors mentioned “Korean market” but which Korea? North Korea or South Korea?
    • Deepen the description of the limitations of conducted research and indicate the trends for further empirical research.
    • To expand the managerial implications in the article.
    • The conclusions section needs to be greatly improved and expanded.
    • Human proofreading, English grammar and spelling correction are also required in order to improve the quality of the manuscript.
    • I would also like to see a well-developed discussion comparing and contrasting solution/results presented in the work with existing work and then a subsection of it presenting contributions to theory/knowledge/literature and followed by a subsection on “Implications for practice”.
    • There are some very strange notations for tables such as: <Table 1>, Table 1., Table 2.,Table 3., <Table 4> and so on, which do not follow the instructions for authors provided by the journal;
    • The sources must be added under each figure.
    • The Questionnaire must be added as appendix.
    • The References section does not follow the instructions for authors provided by the journal. References in the text must be included following the instructions for authors based on the standards of Foods journal.

I consider this article needs to be significantly improved because at this moment it does not meet the academic standards for publication.

Author Response

  • The paper covers a challenging topic of certain interest for researchers and academics.
  • The authors should consider the following recommendations in order to improve the original manuscript:
  • To include the structure of the paper in the Introduction section;.

Response:  According to your point, the flow of the study and the necessity of IPA were mentioned for the purpose of the study. Please see line numbers from 61 to 64 on page 2. We hope this modification adequately addresses your comments. Thank you.

“This study aimed to identify features that make HMRs sold in South Korea attractive to Korean consumers. As a specific research method, the concept and characteristics of HMR were identified, and the current status and trends of the HMR market were investigated.”

  • To include the research hypotheses;
  • There are no research questions so must be included;

Response:  Since IPA is an analysis that identifies the importance and performance of factors, a research hypothesis is not required. Nevertheless, as you suggested, the hypothesis setting of the study is the primary basis for scientifically verifying social phenomena.

Therefore, instead of establishing a hypothesis, the differentiation and necessity of IPA analysis were sufficiently explained with the background of the study in the introduction. We hope this addition adequately addresses your comments. Please see line number 51, and form 56 to 59 on page 2. Thank you.

“What service strategies should the HMR market have for sustainable consumption?”

“Despite HMR-related research achievements in various academic fields, research to understand the characteristics of products is essential in seeking a sustainable service strategy of COVID-19 [25, 25]. The IPA model is used as a significant model for evaluating service quality performance and predicting consumer behavior [26, 27, 28].”

  • There provided very few comments about COVID-19 pandemic and its globally devastating impact. How affects Covid-19 pandemic this research idea and its findings?

Response:  The need to analyze consumer consumption behavior according to the COVID-19 pandemic was mentioned in the abstract. Please see line numbers from 31 to 15 on page 1.

“Moreover, the COVID-19 pandemic has changed consumers' traditional consumption habits and requires new service strategies to cope with sustainable consumption behavior.”

In addition, according to your proposal, the COVID-19 impact was significantly revised in '2. Theoretical Background', '6. Discussion', and '7. Conclusion'. This part will be explained in more detail in the commentary answer below. I appreciate the criticism.

  • It is more than necessary to expand considerably the section "Literature review". The authors also did not provide sufficient evidence on literature review to support the hypotheses. The Literature review section which is practically non-existent being mentioned only a few bibliographic references quite uncorrelated. Authors should take into consideration much more recent publications in the sphere of discussed subject matter, especially studies conducted during the last 5 years.

Response:  According to your suggestion, our research team has significantly revised the theoretical background. The number of references increased from 48 to 75, and most of the references were published within five years.

  • Regarding consumer behaviour framework, I suggest extending the literature section by Also, from the supplier's point of view, HMR's supply system improvement plan and sustainable industrial growth plan should be prepared.including at least the following relevant studies:
    1. Hawaldar, I.T.; Ullal, M.S.; Birau, F.R.; Spulbar, C.M. Trapping Fake Discounts as Drivers of Real Revenues and Their Impact on Consumer’s Behavior in India: A Case Study. Sustainability 2019.
    2. Timpanaro, G.; Bellia, C.; Foti, V.T.; Scuderi, A. Consumer Behaviour of Purchasing Biofortified Food Products. Sustainability 2020, 12, 6297.
    3. Jing, X.; Guanxin, Y.; Panqian, D. Quality Decision-Making Behavior of Bodies Participating in the Agri-Foods E-Supply Chain. Sustainability 2020, 12, 1874.

Response:  As you pointed out, this study requires a practical approach to spreading HMR sustainable consumption. Many of these parts were omitted from the existing manuscript. Thank you very much for introducing good studies related to this study. You helped me a lot.

Many parts have been revised based on the expansion of the perspective you gave me and the referenced proposal.

First, 'Definition of HMR' was revised to 'Characteristics and Consumption of HMR', and HMR consumption behavior and industrial process innovation were described by fully reflecting the research samples proposed by you. We hope this addition adequately addresses your comments. Please see line numbers from 81 to 91 and 94 to 96 on page 2. Thank you.

“Modern people's diet-lifestyle is much simpler and faster than before [31, 32]. In particular, this tendency is stronger for Asians than for Westerners and tends to pursue convenience in the traditional diet [33]. Also, the spread of food delivery culture in pandemics led to the growth of HMR [34]. The changed diet lifestyle can explain and predict HMR consumption behavior as food purchase behavior [35, 36]. Wise consumers seek institutional, scientific, and biologically safe foods [37]. Previously, consumers mainly purchased HMRs at large discount stores but made more purchases online after COVID-19. HMR marketing activities are also primarily carried out on SNS and online to target the Z generation and young couples [38, 39]. Since HMR is a consumer product with a relatively low degree of involvement, innovative supply management such as an e-chain system or quality certification can significantly influence decision-making [40].”

“Nevertheless, HMR has steadily developed to meet consumer needs due to production and distribution technology development.”

Second, in order to supplement the reliability and validity of IPA analysis, the background of previous studies was further added in '2. Attributes of HMRs'. Please see line numbers from 106 to 107, and from 109 to 113 on page 3. We hope this addition adequately addresses your comments. Thank you.

“These economic and social changes make lifestyle seeking simplicity, convenience, and practicality common [46].”

“Diet lifestyle measures consumer behavior that reflects consumers' life values with external factors such as social perspectives, traditions, and safety. Internal factors shall measure the consumption behavior of the product, such as health, hygiene, taste, product quality, freshness, convenience, and information about the product [47].”

Third, along with Amazon's case, the industrial innovation required in the HMR market was emphasized in '2.3. The HMR market'. Please see line numbers from 161 to 163 on page 4. We hope this addition adequately addresses your comments. Thank you.

“Global distribution and logistics companies such as Amazon have recently been competitively expanding their investment in the cold chain to more quickly deliver the fresh food they need at a small quantity and reasonable cost [61].”

  • Authors mentioned “Korean market” but which Korea? North Korea or South Korea?

Response:  According to your suggestion, it was clarified that it was South Korea. Please see line number 61on page 1, and line number 86 on page5. Thank you.

“This study aimed to identify features that make HMRs sold in South Korea”

“The survey data of this study was collected in South Korea.”

  • Deepen the description of the limitations of conducted research and indicate the trends for further empirical research.

Response:  According to your point, the study's limitations and proposals for future research have been faithfully revised. In order to aid in the methodological understanding of the limitations, the effectiveness of IPA was first mentioned in ;4.3 Implicitly derived importance of HMR attributes'. Please see line numbers from 225to 227 on page 8. We hope this addition adequately addresses your comments. Thank you.

“(Modified) IPA has been applied to many studies related to consumer behavior and marketing to explain better the exact degree of consumer awareness [27, 28].”

  • To expand the managerial implications in the article.
  • The conclusions section needs to be greatly improved and expanded.
  • I would also like to see a well-developed discussion comparing and contrasting solution/results presented in the work with existing work and then a subsection of it presenting contributions to theory/knowledge/literature and followed by a subsection on “Implications for practice”.

Response:  As you pointed out, many parts of the discussion and conclusions needed to be revised according to the revision of the theoretical background. Therefore, our research team decided to significantly revise the part, originally one conclusion, into three parts: '5. Discussion', '6. Conclusion', and '7. Limitation and Future Research'.

The context of the revision is as follows.

First, '5. Discussion' describes the theoretical and practical implications of IPA results, and in particular, it was revised by focusing on the industrial response of HMR to changes in consumer behavior. Please see line numbers from 259 to 261 on page 10, 271 to 291, 301 to 305, and 310 to 312 on page 11. We hope this addition adequately addresses your comments. Thank you.

“Many studies using the IPA have made recommendations of service improvement and marketing strategies based on Quadrant I (concentrate here) and Quadrant II (keep up the good work) [66, 67].”

“The creation of the taste of HMR should be considered not only in the social and cultural environment of the market but also in the aesthetic and moral decisions of consumers. As global interest in health and the environment increases, consumption of healthy foods such as vegetables and organic foods increases [68]. HMR should develop premium organic menus that consider vegetarian diets, nutritional selection attributes, and purchasing attitudes. Today's elderly people are defined as new generations, and accordingly, new perspectives and preparations are needed. The most fundamental concern of these generations, whose assets have increased and tastes have diversified, is "health care." Moreover, the starting point of their health care always starts with food. If HMR develops it as an eco-friendly organic food, it can remove the image of instant food and do brand positioning as a sustainable food.

Items corresponding to the price of HMRs were among those requiring intensive improvement. As there are more alternatives that consumers can choose, consumers react sensitively to prices and value cost-effectiveness [20]. The rising consumer Z generation shows more rational and ethical consumption tendencies than in the past [69, 70]. They aim for empirical consumption, not material consumption; they buy the necessary items to a minimum or wander for a large discount even though they can afford it economically [71]. On the other hand, eco-friendly products strive for sustainable consumption at a little more cost. In other words, it is necessary to develop an HMR that can save time and use it practically. It is necessary to find attributes that affect consumer price sensitivity and induce consumers to recognize products and consume reasonably.”

“SNS marketing is helpful for brand positioning for customers by producing content that promotes brands or products [73]. It is good to open cooking classes and introduce new foods through YouTube or SNS. Providing tips to demonstrate how to cook HMR simply or eat it deliciously will help consumers recognize it as friendly and be willing to purchase it.”

“Suppliers should establish their energy efficiency standards and service standards to produce, store stably, and distribute products through cold chains.”

Second, '6. Conclusion' set the future direction of HMR and focused on the spread of sustainable consumption after COVID-19. We hope this addition adequately addresses your comments. Please see line numbers from 329 to 338 on page 12. Thank you.

“Another condition that consumers value in decision-making is sustainability [74]. The number of pollutants emitted in the process of product production or service execution, recyclability after use, and whether toxic chemicals have been replaced with natural ingredients have become essential factors in consumers' purchasing decisions [75]. In addition, food upcycling is emerging as a new trend. Eco-friendly products are being developed made from food by-products that were naturally discarded, such as bean curd and broken rice. Therefore, there is a need for continuous R&D that can satisfy both sustainability and value consumption. In addition, innovative solutions should be developed along with a conscious supply chain, and responsible ESG management should be introduced so that the HMR market system can be transformed into a sustainable way.”

Third, the limitations of this research methodology were described in fact, and subsequent research topics that could be supplemented were proposed in '7. Limitations and Future Research'. We hope this addition adequately addresses your comments. Please see line numbers from 345 to 350 on page 12. Thank you.

“Third, implications for all HMR items in the fourth quadrant of IPA results were not derived, and consumer behavior control and impact variables were not considered. The attributes of HMR are expected to become more diverse as the supply system continues to develop and the food market becomes globalized. Future research should identify problems of HMR consumption from a consumer perspective and find ways to improve services. Also, from the supplier's point of view, HMR's supply system improvement plan and sustainable industrial growth plan should be prepared.”

  • Human proofreading, English grammar and spelling correction are also required in order to improve the quality of the manuscript.

Response:  The deficiencies will continue to be supplemented with Proofreading after correction. Thank you for taking a close look.

  • There are some very strange notations for tables such as: <Table 1>, Table 1., Table 2.,Table 3., <Table 4> and so on, which do not follow the instructions for authors provided by the journal;

Response:  According to your point, all modifications were made according to the MDPI form. Thank you.

  • The sources must be added under each figure.

Response:  According to your point, the source is indicated and the abbreviation in the table is explained in notes.

  • The Questionnaire must be added as appendix.

Response:  I present the questionnaire to you as follows. In fact, all surveys were written in Korean and were conducted using the Google Online Survey Form. However, we will decide later whether to insert the questionnaire as an appendix at the bottom of the paper or not according to your answer and the editor's opinion of the Foods journal. Please understand this.

  1. This is a question about the HMR selection attributes of the following items. Please mark (V) in the corresponding number of items you think about HMR.

Question

extremely

disagree 

<->

extremely

 agree

1.

I value the unique taste of HMR.

â‘ 

â‘¡

â‘¢

â‘£

⑤

2.

I look for HMR with various flavors.

â‘ 

â‘¡

â‘¢

â‘£

⑤

3.

I enjoy HMR's introduction to the new dish.

â‘ 

â‘¡

â‘¢

â‘£

⑤

4.

I prefer the new flavor of HMR.

â‘ 

â‘¡

â‘¢

â‘£

⑤

5.

Buying HMR is cheaper than cooking at home.

â‘ 

â‘¡

â‘¢

â‘£

⑤

6.

HMR should be available for purchase at reasonable prices.

â‘ 

â‘¡

â‘¢

â‘£

⑤

7.

I buy HMR according to efficiency at low cost.

â‘ 

â‘¡

â‘¢

â‘£

⑤

8.

I buy HMR according to cost-effectiveness.

â‘ 

â‘¡

â‘¢

â‘£

⑤

9.

HMR should be easy to prepare

â‘ 

â‘¡

â‘¢

â‘£

⑤

10.

HMR should be simple to cook

â‘ 

â‘¡

â‘¢

â‘£

⑤

11.

HMR should be easy to store.

â‘ 

â‘¡

â‘¢

â‘£

⑤

12.

HMR should be easily available

â‘ 

â‘¡

â‘¢

â‘£

⑤

13.

HMR should be neatly packed

â‘ 

â‘¡

â‘¢

â‘£

⑤

14.

HMR should provide accurate component information.

â‘ 

â‘¡

â‘¢

â‘£

⑤

15.

HMR should accurately label the ingredients.

â‘ 

â‘¡

â‘¢

â‘£

⑤

16.

HMR should indicate the exact country of origin.

â‘ 

â‘¡

â‘¢

â‘£

⑤

B.

Question

extremely

disagree 

<->

extremely

 agree

What is your overall satisfaction with HMR?

â‘ 

â‘¡

â‘¢

â‘£

⑤

  1. Here are some of your general points. Please mark (V) on the corresponding number.

  1. What is your gender?

â‘  Female                         â‘¡ Male

  1. How old are you?

â‘  Under 30        â‘¡ 30–39             â‘¢ 40–49              â‘£ Over 50

  1. What is your occupation?

â‘  Students                      â‘¡ Office             â‘¢ Service           â‘£ Housewife

⑤ Self-employed             â‘¥ Others

  1. How much is your monthly income?

â‘  Under $2,000               â‘¡ $2,000–$3,000              â‘¢ $3,000–$5,000               â‘£ Over $5,000

  1. How many times a month do you buy HMR?

â‘  0–2 times                  â‘¡ 3–5 times       â‘¢ 6–9 times       â‘£ Over ten times

  1. How much do you spend on average when purchasing HMR?

â‘  Under $6       â‘¡ $6–$10         â‘¢ Over $10

  1. This is a question about the HMR selection attributes of the following items. Please mark (V) in the corresponding number of items you think about HMR.

  • The References section does not follow the instructions for authors provided by the journal. References in the text must be included following the instructions for authors based on the standards of Foods journal.

Response:  All references were modified according to the foods journal form. Thank you.

  • I consider this article needs to be significantly improved because at this moment it does not meet the academic standards for publication.

Response:  Most of the criticisms have helped this paper achieve completeness. Our research teams have learned a lot by contemplating and revising your point, and thank you once again for your active process support from MDPI.

With Omicron still actively spreading, I hope you will always be healthy.

Round 2

Reviewer 1 Report

The Authors complied with my requests, but some modifications introduce some further problems.

The Authors replied that the home-meal replacement is a generalized analysis so they decided to not include "South Korea" in the title. This is contradictory with their sentences at lines 80-82 "Modern people's diet-lifestyle is much simpler and faster than before [31, 32]. In particular, this tendency is stronger for Asians than for Westerners and tends to pursue convenience in the traditional diet [33]." stating that territoriality is crucial. So at least in the abstract, it is suggested to cite South Korea.
- the titles of Tables 4 and 5 are different from those reported in the Authors' replies. I assume it was a mistake in the answers.

Minor remarks
- line 87 please add the extended name of SNS
- line 88 please add a reference for "Generation Z"
- FRL introduced at line 113 is used only once at line 126. I suggest removing the abbreviation that unnecessarily burdens the text.
- line 57 the reference results [25, 25]. I feel it should be [25]
- line 229 I suggest changing (=PCC) with "expressed by PCC reported in brackets)
- lines 245-246  "Quadrant III is composed of elements with low satisfaction but of importance and thus of relatively low priority" should e modified into "Quadrant III is composed of elements with low satisfaction and low importance and thus of relatively low priority.

Author Response

The Authors complied with my requests, but some modifications introduce some further problems.

The Authors replied that the home-meal replacement is a generalized analysis so they decided to not include "South Korea" in the title. This is contradictory with their sentences at lines 80-82 "Modern people's diet-lifestyle is much simpler and faster than before [31, 32]. In particular, this tendency is stronger for Asians than for Westerners and tends to pursue convenience in the traditional diet [33]." stating that territoriality is crucial. So at least in the abstract, it is suggested to cite South Korea.

response: At your suggestion, I mentioned the need for hmr in the Korean market in abstract, and the title also added 'in Korea'. The changed title is as follows.

Sustainable Home Meal Replacement (HMR) Consumption: Exploring Service Strategies in Korea Using a Modified Importance-Performance Analysis

- the titles of Tables 4 and 5 are different from those reported in the Authors' replies. I assume it was a mistake in the answers.

Response: I checked again, but there was nothing wrong.

Minor remarks

- line 87 please add the extended name of SNS

- line 88 please add a reference for "Generation Z"

response: A supplementary explanation is described as follows.

“HMR marketing activities are also primarily carried out on SNS and online to target the Z generation and young couples who are highly dependent on mobile purchases [38, 39].”

- FRL introduced at line 113 is used only once at line 126. I suggest removing the abbreviation that unnecessarily burdens the text.

response:The abbreviation of FRL was deleted and changed to 'diary life' in another quote.

- line 57 the reference results [25, 25]. I feel it should be [25]

response: Thank you. I have corrected it.

- line 229 I suggest changing (=PCC) with "expressed by PCC reported in brackets)

- lines 245-246  "Quadrant III is composed of elements with low satisfaction but of importance and thus of relatively low priority" should e modified into "Quadrant III is composed of elements with low satisfaction and low importance and thus of relatively low priority.

response:Thank you. I have corrected it.

Thank you very much for your hard work until the end.

Reviewer 2 Report

The original manuscript has been improved. The authors followed the recommendations included in the previous review report so that the quality of their research article has increased. I also appreciate the effort of the authors in this regards.

Author Response

Thank you for your comment based on your professional opinion.

It was a great help to improve the completeness of the study.

The English language and grammatical errors pointed out will be corrected along with proofreading.

Thank you.